# Changes in Community Composition and Functional Traits of Bumblebees in an Alpine Ecosystem Relate to Climate Warming

**DOI:** 10.3390/biology12020316

**Published:** 2023-02-16

**Authors:** Victor Sebastian Scharnhorst, Katharina Thierolf, Johann Neumayer, Benedikt Becsi, Herbert Formayer, Julia Lanner, Esther Ockermüller, Alina Mirwald, Barbara König, Monika Kriechbaum, Harald Meimberg, Philipp Meyer, Christina Rupprecht, Bärbel Pachinger

**Affiliations:** 1Institute for Integrative Nature Conservation Research, University of Natural Resources and Life Sciences Vienna, 1180 Vienna, Austria; 2Freelance Entomologist, Obergrubstraße 18, 5161 Elixhausen, Austria; 3Institute of Meteorology and Climatology, University of Natural Resources and Life Sciences Vienna, 1180 Vienna, Austria; 4Department of Ecology, University of Innsbruck, 6020 Innsbruck, Austria; 5Biology Centre of the Upper Austrian State Museum, 4040 Linz, Austria

**Keywords:** mountain ecosystems, ectothermic species, range shifts, Anthophila, *Bombus*, species temperature index (STI), collection work, historical data, Tyrol, Austria

## Abstract

**Simple Summary:**

Climate warming affects diversity, community composition, and spatial distribution of several plant and invertebrate species. Bumblebees in alpine ecosystems are particularly exposed to climate change due to even stronger warming compared to the global mean. To investigate the effects of climate warming, we sampled bumblebees along mountain slopes, compared the records to historical data from 1935 and 1936 and related our findings to climate models. We found that bumblebee species communities differed significantly between the two sampling periods. Our analyses showed that rising temperatures in the spring were the most plausible factor explaining this shift in community composition. Moreover, the recent bumblebee data showed significantly lower species diversity compared to the historical records. For example, the number of cuckoo bee species (socio-parasitic bumblebees that use the nests of other species for reproduction) was significantly lower compared to the historical data. Even though we did not detect more warmth-loving species, recent communities showed an increase in species that can deal well with variable climatic conditions. We conclude that the composition and functionality of bumblebee communities in the study area have been significantly affected by climate warming, with land use and vegetation changes likely playing an additional important role.

**Abstract:**

Climate warming has been observed as the main cause of changes in diversity, community composition, and spatial distribution of different plant and invertebrate species. Due to even stronger warming compared to the global mean, bumblebees in alpine ecosystems are particularly exposed to these changes. To investigate the effects of climate warming, we sampled bumblebees along an elevational gradient, compared the records with data from 1935 and 1936, and related our results to climate models. We found that bumblebee community composition differed significantly between sampling periods and that increasing temperatures in spring were the most plausible factor explaining these range shifts. In addition, species diversity estimates were significantly lower compared to historical records. The number of socio-parasitic species was significantly higher in the historical communities, while recent communities showed increases in climate generalists and forest species at lower elevations. Nevertheless, no significant changes in community-weighted means of a species temperature index (STI) or the number of cold-adapted species were detected, likely due to the historical data resolution. We conclude that the composition and functionality of bumblebee communities in the study area have been significantly affected by climate warming, with changes in land use and vegetation cover likely playing an additional important role.

## 1. Introduction

Global climate change has led to a worldwide increase in temperature [1] that is significantly affecting many ecosystems [2]. In addition to a gradual warming of annual mean temperatures, extreme events such as heat waves are expected to occur more frequently and last longer, while precipitation patterns will also change, such as more frequent and intense heavy rainfall events [1]. As a result, glacier cover and volume are decreasing, snow cover is declining, permafrost is thawing [3], and tree boundaries are shifting [4,5]. This trend is reflected in changes in species distribution, phenology, and physiology [6]. Other effects of climate change include phenological or spatial imbalances in pollinator networks due to differences in the temporal and spatial evolution of species, as well as increasing pressure from diseases and parasites [7,8].

Alpine ecosystems are particularly affected because temperatures in these areas are rising twice as fast as the global average [9]. Climate warming, among other factors, affects distribution patterns and often drives species to higher elevations and latitudes [10,11,12]. This has resulted in substantial upward range shifts averaging 6.1 m per decade (or kilometer per decade toward the poles) and a long-term loss of species richness at all elevations, although diversity has temporarily increased at cooler and higher elevations [13,14]. When species have already reached their distributional and ecoclimatic limits, the negative environmental effects are even more drastic [15]. Colonization of other areas where climate change creates conditions that fall within the species’ climatic range is less likely [16]. Because of the negative interplay between changing climatic conditions and human land-use change [17,18], species in alpine ecosystems deserve great attention in conservation work [19].

For example, climatically suitable areas for about 77% of European bumblebee species are forecasted to shrink, with more than one-third of the total number of species at risk of extinction [20]. Bumblebees provide important pollination services to many wild flowering plants and crops [21,22], making them keystone species to the maintenance of many ecosystems [23]. Because their special thermoregulatory abilities and insulating dense hair cover make them better adapted to cold climates than other flower visitors, they are the most important pollinators in alpine ecosystems and form particularly species-rich communities [22,24]. Bumblebees have been able to recolonize areas that were uninhabited during the last ice ages [25] and are now found in some of the highest elevation and northernmost ecosystems [26]. Since the establishment of a colony depends on a single female individual in spring, bumblebee community composition is particularly influenced by spring temperatures of past climatic conditions [27,28].

Understanding the ecological niche of a species requires consideration of species-specific characteristics and their dependence on environmental factors [29]. Bumblebee species have different habitat preferences and vary in their altitudinal distribution [20]. While many species are generalists with respect to floral resources, nesting habitats, and climatic preferences, e.g., *Bombus pascuorum*, some species are highly specialized [30]. For example, *B. gerstaeckeri* visits only a single plant genus and feeds exclusively on pollen from two *Aconitum* species. Moreover, cuckoo bee species are obligate social parasites that use the nests of other species for reproduction; they generally occur in lower abundances and are highly restricted to the range of their host species [22]. Other species occur only at high alpine elevations, such as *B. alpinus*, which may forage at ambient temperatures near freezing [27,31]. Accordingly, altitudinal range shifts caused by climate warming become threatening when species-specific pollen resources or host populations disappear, or when cold-adapted species cannot migrate to higher altitudes because they already inhabit the highest elevations [32,33]. Therefore, specialized bumblebee species and those with low dispersal abilities are more sensitive to climate change and rising temperature than widespread, generalist species [34,35] that find new pollen, nectar, and nesting sources more easily [36]. This trend could lead to a dominance of generalist species with greater dispersal ability, reproductive potential, and ecological generalization [30,37]. Thus, shifts in species communities toward cooler areas could induce a turnover from cold-adapted to more heat-tolerant and generalist bumblebees at the local scale [38,39]. The consequences for biodiversity and ecosystem functionality could be severe, as generalists may not be able to compensate for the loss of specialized species, e.g., in terms of pollination services [40]. However, given the different extinction debt related lag times of specialist and generalist species [41], generalists are likely to be increasingly affected as habitat quality further declines [42].

To attribute observed changes in bumblebee diversity, community composition, and spatial distribution to climate warming, historical data collections prior to the onset of significant warming are of great importance. In-situ trait response tests are key for understanding the adaptive capacity of bumblebees in terms of physiology, morphology, behavior, phenology, and dispersal [43]. In this study, we investigate the composition of the bumblebee community in the Kalsbach Valley in Austria and compare our results with records from the 1930s [44,45] between 1100 and 2899 masl (meters above sea level). Comparable historical collections of bumblebees exist, for example, in Norway [46]. The records of these surveys were generated from both Løken’s own collections and other sources, and have since been digitized and made available on Artskart (https://artskart.artsdatabanken.no/, accessed on 5 January 2023). Nonetheless, working with historical data requires an approximation of one’s own methods with those of the time [39]. Considering the above, we aimed to examine changes in (i) bumblebee community composition, (ii) bumblebee species diversity, and (iii) the distribution of community-weighted means of functional traits, including in a species temperature index (STI). We hypothesized that (1) bumblebee communities can be separately clustered by elevational levels and differ significantly between sampling periods; (2) changes in community composition are related to increased mean annual spring temperatures; (3) species diversity and distributions of several functional traits differ significantly between sampling periods; (4) the community-weighted mean of the STI is significantly higher nowadays compared to historical records.

## 2. Materials and Methods

### 2.1. Study Area

All bumblebee surveys were conducted in the municipality of Kals am Großglockner (180.5 km²) in the East-Tyrol province in Austria. The region is located in the Eastern Alps, and two-thirds of the municipality area is part of the Hohe Tauern National Park, which was established in 1982. The elevation of the study area ranges from 1100 masl (meters above sea level) to 3100 masl. Since the study area spans four elevational levels (montane, subalpine, alpine, nival), it includes diverse habitats with mountain spruce and pine forests (tree boundary: about 2000 masl), hay meadows, alpine pastures, dwarf shrubs, and scree vegetation [47]. The region belongs to a temperate alpine climate zone with a long cold season, moderate precipitation in winter, and medium precipitation in summer [48]. In 2020, the mean annual temperature in Kals was 6.1 °C with a total annual precipitation of 981 mm [49]. 

### 2.2. Bumblebee Data Survey

We used the Pittioni Bee Collection for the historical reference data [44,45], a well-documented bee collection predominantly from Austria, held by the Natural History Museum, London. The original index cards are available online (http://pittioni.myspecies.info, accessed on 5 January 2023). It is one of the most accurate data collections, considering that quantitative surveys were not common at that time. The historical bumblebee data were sampled by Bruno Pittioni in Kals am Großglockner between 1935 and 1937 at various sampling locations (depicted in Figure 1). However, only a few scattered records are available from 1937, and it can be assumed that Pittioni did not conduct extensive sampling after 1936. Further, it is important to acknowledge that the historical sampling carried out by Pittioni in 1935 and 1936 did not consist of revisiting the same locations over two years, but rather different locations were sampled in those years. Most of Pittioni’s bumblebee records included information on species, sex, abundance, location, elevation, and date, with additional information on flowering plants visited. As Pittioni assigned samples to subspecies and used synonyms that are no longer accepted, the index cards were digitized and species names transcribed into the current nomenclature following Amiet [21]. Since the index cards do not contain coordinates for the origin of the samples, the records were georeferenced through locality description and elevation on index cards along with historic maps. Despite fairly extensive documentation, the sampling strategy of Pittioni is unknown. However, a semiquantitative sampling procedure can be assumed, as he reports in his publication that he selected specific sampling areas and was following an elevational gradient [44].

We conducted bumblebee surveys between 20 July and 22 August 2020 at several sampling locations (Figure 1). Sampling areas and dates were selected based on information from historical data collections [45] to increase the comparability of both time periods. Bumblebees were sampled at Kalsbachtal (hereafter KBT), Lesachtal (LT), Ködnitztal (KT), Foledischnitz (FS), Teischnitztal (TS), Dorfertal (DT), Raseggbachtal (RS), Kals-Matreier-Törl (KMT), Schönleitenspitze (SL), Figerhorn (FH), and at the base of the mountain Großglockner (GG). We standardized the sampling procedure and systematically collected bumblebees in 4-m-wide transects, using hiking trails and forest roads as routes that were explored in historical surveys. We started a new transect when the habitat changed, a new elevation level began (using 100 m increments), or a distance of 1000 m was reached. The total length of each transect was between 200 and 1000 m. In addition, we observed 2 × 2 m plots for 15 min at selected coordinates at different elevations and habitats studied in historical surveys. In total, we sampled 116 transects (62 km) between 1100 and 3100 masl and observed 18 plots. In general, we conducted mapping throughout the day as it was not possible to map in some areas for only a specific time of day due to the length of some transects. The main mapping time was between 9am and 5pm; sometimes activity was observed even after 5pm. If possible or when changing study areas, we attempted to avoid sampling during noon as bumblebees appeared to be less active (subjective experience). Our mapping was primarily guided by environmental conditions rather than a specific time of day. All observed bumblebees were recorded and, when possible, identified in the field using the key of Gokcezade et al. [51] or transferred to the laboratory and identified according to Amiet [21]. The latter specimens are preserved in the collection of the Institute for Integrative Nature Conservation Research, University of Natural Resources and Life Science, Vienna. A small number of bumblebee specimens were not able to be identified to species level in the field due to the specimens flying away or being uncatchable. These specimens were recorded as *Bombus sp.* and were excluded from any subsequent analysis.

### 2.3. Climate Data

We evaluated temperature data for the 30 years prior to each sampling date (1906–1935 and 1991–2020, see Figure 2) by calculating four temperature indices with high spatial resolution for the study area: annual mean temperature, mean temperature of the summer half-year (April–September), mean temperature of the winter half-year (October–March), and mean spring temperature. Due to the high elevation of the study area, we did not use the meteorological standard definition of spring including the months March, April, and May. Instead, we used the period March, April, May, and June. This period is better linked to the end of snowmelt and the start of the vegetation season within the study area. The annual values of these indices were averaged over the two periods.

Two data sets were used as the data basis: The gridded observational dataset SPARTACUS [52] is available for Austria and some adjacent regions in neighboring countries with a spatial resolution of 1×1 km and a daily temporal resolution, and was used directly to calculate temperature indices for the period 1991–2020. The long-term dataset HISTALP [53] provides homogenized instrumental data back to the 18th century with a grid resolution of 5 × 5 km and monthly timesteps, for the domain of the European Alps. It was used to evaluate historical temperatures for the period 1906–1935.

The HISTALP and SPARTACUS data were combined to achieve a dataset with the spatial resolution of SPARTACUS and the monthly timesteps and long-term temporal availability starting in the 18th century. First, HISTALP data was regridded to the SPARTACUS grid using the Earth System Modeling Framework’s (ESMF) ‘Higher-order patch recovery’ or ‘patch’ interpolation method implemented in the xESMF Python package [54]. This method works by constructing multiple polynomial patches from cells surrounding the source cell. The interpolated value at the destination point is the weighted average of the values of the patches at that point [55,56]. After regridding, a linear regression model was fitted for each month and pixel for the period 1961–1990, which is available in both datasets, with SPARTACUS as the dependent variable. The resulting coefficients were applied as bias correction to the HISTALP data for the period previous to 1961. From that point forward, monthly averages of the original SPARTACUS data were used.

For a more detailed analysis of the study area, the four temperature indices were downscaled to a 10 × 10 m resolution using the local vertical temperature gradient (−6.68 K/1000 m). Residuals were calculated by subtracting the height dependency (temperature gradient * elevation [1 km]) from temperature values. Those residuals were again interpolated to a 10 × 10 m grid using the ‘patch’ method described above. A 10 × 10 m digital elevation model of the Tyrolean Spatial Information System (TIRIS) was used to add back the height dependency (temperature gradient * elevation [10 m]) to the downscaled residuals to create the final temperature indices fields (see Figure 3). Microclimatological effects such as the cooling of the glacier areas are not considered within this study. Mean values of the four indices were calculated for the elevation zones corresponding to bumblebee data from 1935/1936 and 2020. All calculations were performed with the NumPy (v1.21.5), xESMF (v0.3.0), and Xarray (v0.20.1) Python packages.

### 2.4. Bumblebee Species Traits

Trait values of all bumblebee species assessed in this study were taken from recent literature on bumblebee ecology [21,57,58,59]. Some additional traits were based on the expert assessment of Johann Neumayer. All traits that were used in the subsequent statistical analysis and a description of the bumblebee trait categories are provided in the Appendix A (Table A1 and Table A2). The species *B. lucorum* and *B. cryptarum* were assigned to the *Bombus lucorum/cryptarum* complex because these species could not be distinguished with certainty in the 1930s and trait values were averaged between the two species.

### 2.5. Statistical Analyses

#### 2.5.1. Processing of Data Records

All analyses were conducted in R Version 4.0.3 [60] using the associated packages in “tidyverse” [61]. We used only historical bumblebee records from 1935 and 1936, as there was no comprehensive sampling carried out in 1937. Historical records with an uncertainty of elevation >500 m or without information about recording dates were excluded from the analysis (e.g., all data from RT). We also excluded records if we suspected that data belonged to a single excavation of a nest (e.g., only one species recorded for a location but in a high number of individuals). In addition, records of *B. mesomelas* and *B. campestris* were excluded as no information on the number of individuals of these species was provided for 1935/1936. To allow comparison of the same elevations between the 1935/1936 and 2020 data, we evaluated bumblebee records only from 1100 to 2899 masl.

For subsequent analyses, we pooled all bumblebee species data per year (e.g., 1935) at a distinct sampling site (e.g., KBT) and elevation (e.g., 1900–2000 masl); hereafter, they are referred to as sampling units. To integrate results from climate data, we averaged the upper and lower elevation boundaries of each sampling unit (e.g., “1935#KBT#1950MSL”; MSL = mean sea level). Many historical records contained only imprecisely defined elevations (e.g., 1900–2400 masl). To reflect the historical data structure as best as possible, these elevations were replicated for the data from 2020 and sampling units pooled accordingly. If several sampling units from 1935/36 were redundant for a sampling site, the data were also pooled (e.g., 1900–2000 masl and 1800–2000 masl).

To avoid bias in species abundance from the historical data, we converted the data into a binomial matrix (presence/absence) of species and sampling units that was used throughout the analysis. To gain information on ecological similarity of species communities of the sampling units, we performed a hierarchical cluster analysis. First, a Sørensen dissimilarity matrix was created using the function “vegdist” (method = “bray”) in the package “vegan” [62]. The function “hclust” (method = “ward.D”) in the package “factoextra” [63] was then used to extract groups (Bray-Curtis distance = 1.5) based on the obtained dissimilarity matrix using Ward’s minimum variance method. In the following analyses, the gained groups were then compared among the two sampling periods (hereafter referred to as “cluster groups”).

#### 2.5.2. Analyses on Community Composition

To investigate bumblebee community composition of different cluster groups and relevant impacts of climate factors, a constrained ordination analysis was performed. Based on the Sørensen dissimilarity matrix, as described above, we performed a canonical analysis of principal coordinates (CAP) using the function “capscale” in the package “vegan” [62]. Using the function “ordistep” (direction = “both”) and subsequent ANOVAs, we tested the significance of variables such as elevational level of sampling units, cluster group, climate data (e.g., spring temperature, see 2.3), and sampling period against an unconstrained null model and selected the best model obtained (ANOVA: *p* < 0.05). The R²-value of the CAP was calculated with the function “RsquareAdj”. Constraint variables and mean elevation of sampling units were fitted as vectors into the reduced ordination space via the function “envfit”. Further, we displayed the position of each bumblebee species in the reduced ordination space. Using the function “adonis” (permutations = 999) and the Sørensen dissimilarity matrix, we performed a PERMANOVA analysis to test the influence of the aforementioned variables on bumblebee community composition. Furthermore, we used the function “betadisper” and subsequent ANOVAs to control for multivariate homogeneity of groups dispersions. All tests were controlled for multiple comparisons.

#### 2.5.3. Species Diversity Metrics and Sample Coverage

Bumblebee species diversity and sample coverage of the cluster groups were compared with a rarefaction-extrapolation analysis using the function “iNEXT” (datatype = “incidence_raw”) in the eponymous package [64]. For each cluster group, we calculated three diversity metrics expressed in Hill’s numbers, representing a unified family of diversity indices [65]: rarefied species richness (q = 0), Shannon’s diversity index (q = 1), and inverse Simpson’s index (q = 2). Estimates for each diversity metric were presented as functions of the number of sampling units because species diversity metrics are sensitive to sample completeness. For sample sizes smaller or larger than the actual number of sampling units per cluster group, estimators were calculated using rarefaction and extrapolation, respectively, and curves were plotted for each diversity metric. Estimators for each Hill number were extrapolated to a maximum of two times the actual number of sampling units according to Chao et al. [66], and 95% confidence intervals for each diversity metric curve were obtained by bootstrapping. For a qualitative comparison, we also plotted the sample coverage for each cluster group. Rarefied estimates and confidence intervals for each diversity metric, equal to two times the least common number of sampling units according to Chao et al. [66], were then extracted using the “estimateD” function. The values obtained were compared among the cluster groups and tested for significant differences with Kruskal-Wallis Rank Sum Tests using the function “kruskal.test”.

#### 2.5.4. Functional Trait Analyses

To evaluate functional composition of bumblebee species communities of the different cluster groups, we compared the distribution of twelve distinct ecological traits (see Table A1 and Table A2). For each trait, community weighted means (CWM) were calculated with the function “functcomp” in the package “FD” [67,68]. The calculated values thereby represent the average of the respective species trait of each sampling unit in relation to the relative abundance of the distinct species [69]. The traits comprised worker proboscis length in millimeters, worker body size in millimeters, nesting behavior (aboveground or belowground), social parasitism (cuckoo bee species: *Bombus bohemicus*, *B. barbutellus, B. campestris, B. flavidus*, *B. quadricolor*, *B. rupestris,* and *B. sylvestris*), a species temperature index (STI) according to Rasmont et al. [20], habitat preference (open areas or forests/edges/transitional), microclimatic preference (wide or cold), and macroclimatic preference (indiscriminate or alpine). Values for all traits were scaled and centered prior to CWM calculation. Using the obtained CWM values, we then compared the distribution of species traits among groups of elevational clusters and sampling periods using Kruskal-Wallis rank sum tests with the “stat_compare_means” function in the “ggstatsplot” package [70].

## 3. Results

### 3.1. Climate Warming

Averaged over the study area, spring temperatures (March–June) increased by 1.76 K between the two study periods 1906–1935 and 1991–2020. Mean spring temperatures remained stable until the 1970s and increased rapidly since 1980 (see Figure 2). The calculated vertical temperature gradient for the study area is 0.668 °C per 100 m. The equivalent mean spring temperature is found to be 257 m higher on average in the 1991–2020 period compared to the 1906–1935 period. Thermal conditions shifted accordingly about 150 m per K.

**Figure 2 biology-12-00316-f002:**
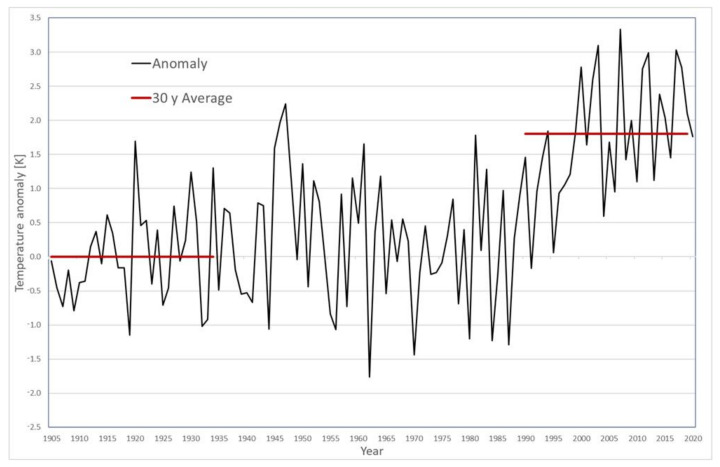
Anomaly of the spring (March—June) temperature (black) at the target area, relative to the period 1906 to 1935. The two red horizontal lines give the 30-year-average spring temperature for the periods 1906 to 1935 and 1991 to 2020. The averages of the two periods differ significantly (*p*-value < 0.01).

In Figure 3, the spatial distribution of spring temperatures in the study area is shown. While they averaged at about 0 °C in the 1906–1935 period, the area mean of spring temperatures in the 1991–2020 period is ~1.7 °C. In the years since the historical period, the 0°C line moved upwards into the formerly glaciated areas of the 1930s (marked with red dashed outlines). The current glacier extent (data from 2012) is shown as the white area. In 1937, 26.8 km² of the study area was glaciated, and glaciers reached down to an elevation of 2300 masl. In 2012, only 8.4 km² of glaciers were left, and the lowermost points are at 2600 masl. This corresponds to a reduction of the glaciated area by approximately 70% compared to 1937.

**Figure 3 biology-12-00316-f003:**
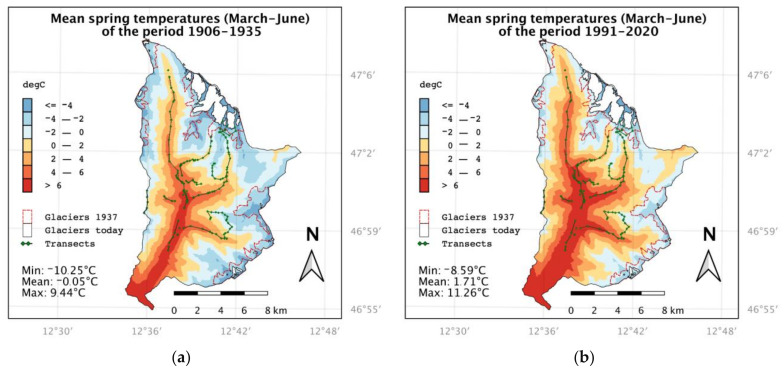
Mean spring temperatures (March–June) of the periods 1906–1935 (**a**) and 1991–2020 (**b**) in Kals am Großglockner. Historic (red—[44]) and recent (white—[50]) glacier cover as well as transects of 2020 (green) are shown on the map. In the lower left, the minimum, mean, and maximum spring temperatures of the study area are noted.

### 3.2. Bumblebee Data

The historical records from 1935 and 1936 comprised a total of 29 species while only 22 species were recorded in 2020. Six species were only found in 1935 and 1936: *Bombus bohemicus*, *B. flavidus*, *B. jonellus*, *B. quadricolor*, *B. subterraneus*, *B. campestris,* and *B. sylvestris*.

### 3.3. Cluster Analysis and Bumblebee Community Composition

Cluster analysis of sampling units revealed two distinct groups (Figure 4a) that corresponded to bumblebee communities at median elevations of 1550 masl (brown, A) and 2150 masl (blue, B). Groups A and B were divided into historic communities from 1935 and 1936 and recent communities from 2020, yielding four distinct cluster groups (see Table A3 for detailed information on species distribution).

Stepwise model selection revealed that bumblebee community composition was best represented in a reduced ordination space by modeling mean annual spring temperature and sampling year as constraint variables (composition ~ mean temperature March–June + sampling year; ANOVA, df = 2 (33), sum of squares = 1.972 (4.377), F = 7.435, *p*-value < 0.001; total R² of CAP ordination = 36.367%). Thus, the mean annual spring temperature explained bumblebee community composition significantly better (*p*-value < 0.05) than the other variables tested, such as mean elevation, mean annual temperature, or mean winter temperature (October–March). Subsequent PERMANOVA analysis showed that mean annual spring temperature (df = 1 (22), sum of squares = 1.527 (1.676), F = 20.048, R² = 30.979 %, *p*-value = 0.001) as well as sampling period (df = 1 (22), sum of squares = 0.412 (1.676), F = 5.414, R² = 8.365 %, *p*-value = 0.002) significantly influenced bumblebee community composition of the sampling units. Other variables such as the sampling location did not have a significant influence (*p*-value > 0.1) when tested together with the aforementioned variables. Accordingly, data points referring to the four different cluster groups (colors: A = brown, B = blue; shapes: 1935/1936 = circles, 2020 = triangles; Figure 4b,d) were clearly separated from each other. Sampling units located in the positive direction of the first ordination axis (CAP1) correlated with higher spring temperatures (vector “T_mean_March_June”) and thus lower elevations and vice versa in the negative direction with lower spring temperatures and thus higher elevations. In addition, data points in the negative direction of the second ordination axis (CAP2) correlated with the sampling period 2020 (“year” vector), and inversely, units in the positive direction correlated with the period 1935/1936. Homogeneity of groups dispersion tests showed no significant differences in variability of bumblebee composition among the four cluster groups (df = 3 (32), sum of squares = 0.086 (0.612), F = 1.496, *p*-value = 0.234), mean annual spring temperature (df = 26 (9), sum of squares = 0.292 (0.612), F = 1.651, *p*-value = 0.219), or sampling year (df = 2 (33), sum of squares = 0.039 (0.754), F = 0.844, *p*-value = 0.439). The position of the different bumblebee species in the reduced ordination space showed that *B. pascuorum*, *B. pratorum,* and *B. hortorum* were particularly associated with high mean spring temperatures and samples from 2020 (Figure 4b,c). The position of *B. soroeensis, B. sichelii, B. lucorum/cryptarum complex,* and *B. pyrenaeus* correlated, in particular, with samples from 2020 and lower temperatures, while the position of *B. ruderarius, B. monticola, B. mendax, B. bohemicus, B. sylvestris,* and *B. flavidus* correlated more strongly with samples from 1935/1936.

### 3.4. Bumblebee Species Diversity

Rarefaction-extrapolation analysis showed that species richness estimates (Hill number q = 0) were significantly higher for 2020 cluster groups compared to 1935/1936 (Kruskal-Wallis test: chi-squared = 3.857, df = 1, *p*-value = 0.049, Figure 5b). The same results were obtained for the Shannon’s diversity index estimates (q = 1, Figure 5c) and inverse Simpson’s diversity index (q = 2, Figure 5d) when comparing cluster groups between 1935/1936 and 2020. Sample coverage was, on average, > 90% for all four cluster groups at an extrapolated/interpolated number of 10 sampling units (equal to two times the least common number of sampling units, Figure 5a).

### 3.5. Distribution of Species Functional Traits

Analyses of the distribution of community weighted mean (CWM) of bumblebee species traits revealed that the proportion of socio-parasitic species was significantly lower in 2020 compared to 1935/1936 in groups A (Kruskal-Wallis test: *p*-value = 0.039, Figure 6e) as well as B (*p*-value = 0.029). In addition, compared to 1935/1936, the 2020 bumblebee communities that belonged to group B had a significantly shorter Proboscis length (*p*-value = 0.023, Figure 6b) and showed a strong trend towards smaller worker body size (*p*-value = 0.059, Figure 6a) and an indiscriminate macroclimatic preference (*p*-value = 0.082, Figure 6k). On the other hand, the proportion of species with a forest/edge/transitional habitat preference (*p*-value = 0.041, Figure 6g), a wide microclimatic preference (*p*-value = 0.017, Figure 6i), and an indiscriminate macroclimatic preference (*p*-value = 0.041, Figure 6k) was significantly higher in the 2020 communities of cluster A compared to those of 1935/1936. The 2020 communities of cluster A also showed a strong trend towards a higher proportion of aboveground nesting bumblebees (*p*-value = 0.068, Figure 6c) compared to 1935/1936. No significant differences were found for other species traits such as community STI or preference of cold microclimates or alpine macroclimates.

## 4. Discussion


Evaluating historical data sources and replicating their sampling methods are key to understanding shifts in species diversity and functional composition associated with major environmental changes such as climate warming. In our study, we combined climate model results with analyses of bumblebee community composition, species diversity, and functional traits in East Tyrol. By replicating historical bumblebee collections from 1935/1936 at the original study sites, we were able to gain insight into community changes that can be directly linked to local climate warming in alpine ecosystems. We further illustrated the great potential of using historical sources as reference data prior to the onset of environmental changes but also showed their limitations due to inaccurate data curation compared to modern ecological research. 

Bumblebee communities clustered according to their elevational distribution and sampling period, indicating a significant species turnover closely linked to increased spring temperatures (Figure 4). Our results demonstrated a significant loss of species diversity at all elevations compared to the historical data (Figure 5). This was particularly linked to the disappearance of species specialized in a socio-parasitic life cycle. Further analyses of functional composition were dichotomous: Communities at lower elevations showed a significant shift towards species characterized by indiscriminate micro- and macroclimatic preferences (Figure 6). On the contrary, there were no positive shifts in community weighted means of a species temperature index or significant loss of species preferring cold and alpine climates. Our results show that climate warming leads to ecological generalization and net loss of bumblebee alpha and beta diversity in alpine ecosystems. Other factors such as changes in land use and vegetation cover and extreme events such as heat waves and drought also affect bumblebee communities; however, we were unable to examine this relationship in this study due to limited data availability. In addition, our conclusions should be considered in light of the limited time frame, as the data were collected over only one- and two-year sample periods.

### 4.1. Cluster Analysis and Bumblebee Community Composition

Bumblebee communities were divided into four distinct cluster groups according to their distribution along the elevational gradient and sampling period (Figure 4). The mean altitudes of the cluster groups roughly classify the communities into below (mean altitude of 1550 masl, group (A)) and above the tree boundary (mean altitude of 2150 masl, group (B)) in the study area. This observation was expected, as bumblebee species have different preferences for open alpine meadows or montane habitats that also comprise forests, edges, or transitional habitats [20,22]. Thus, occurrence in different habitats is strongly associated with the species temperature index and available nectar and pollen resources. We also demonstrated significant turnover in species communities when comparing historical and current sampling years. Here, PERMANOVA analysis and stepwise model selection indicated that mean annual spring temperature explained these differences in bumblebee composition significantly better than the other climate variables tested or mean elevation level alone. Therefore, our study was able to demonstrate a direct relationship between bumblebee community shifts and spring temperature anomalies corresponding to climate warming. 

The computed position of bumblebee species in the two-dimensional reduced ordination space is in line with their climatic preference [20,22], supporting the evidence of our results. Species such as *B. pascuorum, B. hortorum,* and *B. pratorum* are adapted to lower montane habitats and correlate with warm spring temperatures. Others, such as *B. pyrenaeus* and *B. sichelii*, prefer cooler climates and higher alpine elevations and were positioned accordingly in the ordination. We also observed a differentiation of species between sampling years. While *B. ruderarius, B. monticola, B. mendax, B. bohemicus, B. sylvestris,* and *B. flavidus* were typical for historical sampling units, 2020 communities were more characterized by *B. pascuorum, B. pratorum, B. soroeensis, B. sichelii, B. lucorum/cryptarum* complex, and *B. pyrenaeus*. To understand these species distinctions in an ecological context, diversity changes and trait specific responses were examined and discussed below.

### 4.2. Bumble Species Diversity and Functionality

Species richness of bumblebees (Hill’s number q = 0) decreased significantly compared to the historical reference period at both elevational levels in the study area, and sample sizes of the different cluster groups were comparable (Figure 5). These declines were also significant for the Shannon index (q = 1), which considers the number of rare species, and the inverse Simpson index (q = 2), which gives more weight to community evenness. A general decline in bumblebees over the last decades has been documented in previous studies [71], and diversity loss is one of the many consequences of climate change [72] which also applies to our study. However, declines in bumblebee diversity have also been attributed to land use changes [17], particularly agricultural intensification [73,74]. Indeed, a loss of high-quality alpine pastures due to abandonment and usage competition with human activities such as skiing and a decline of grasslands due to construction activities can be observed in the study area [32,75]. This has also led to increasing scrub encroachment and more wooded areas. Moreover, agricultural land in the valley is now mainly used as grassland with long grass and fewer flowers. As bumblebees prefer a mixture of pastures and diverse crop types, it provides lower quality habitats that have a negative impact on species diversity and functionality [76]. Thus, a combination and interaction of climate and land use change [17], including forest expansion [77,78], is the likely reason for the decline in local diversity, changes in community composition, and an increased dominance of a few bumblebee species.

However, it is important to note that while changes in land use and vegetation cover likely also play a role in bumblebee populations, they could not be investigated in our study due to the limitation of not having reliable historical data. Moreover, bumblebee species and their abundance and dominance can vary for a range of reasons, including climatic factors such as previous year heat waves and drought [28], temperature and rainfall that impact mortality and fertility directly and indirectly through changes in flower resources [16], and changes in land use and resource availability [17,24,73,79]. Bumblebee density also affects the opportunities for detection, and weather within a season can have an impact on abundances, with lower abundance after hot days and higher after rainy days [18]. The validity of the conclusions drawn from our study must be considered in the context of these factors, as well as the limitations of our one-season survey in 2020, compared to the two-year data collected in the 1930s. Here, it is significant to note that the historical sampling conducted by Pittioni did not involve revisiting the same locations over the two years, but rather different locations were selected for sampling in each year. This is relevant because bumblebee populations can fluctuate strongly from year to year [24,80], and the temporal range represented by a one- or two-year survey is limited [39].

Our analyses on the distribution of functional traits showed that the diversity loss was closely linked to a significant decrease in species with a socio-parasitic life cycle (Figure 6). Many cuckoo bees such as *B. bohemicus, B. flavidus, B. quadricolor, B. sylvestris,* and *B. campestris* were completely absent for the 2020 data (Figure 6e). Accordingly, there was also a strong association between positions of cuckoo bees and the 1935/1936 group in the CAP ordination biplot (Figure 4c). Because these species depend on their hosts to reproduce and are directly dependent on the stability of their populations, they are one of the first to respond to disturbances [22,81]. This is also expressed in a trend towards smaller body size as these species are comparably large (Figure 6a). Species with long proboscises, which often have special floral preferences [24], *B. mendax*, also decreased significantly in alpine habitats (Figure 6b). In addition, generalist species such as *B. pascuorum* and *B. pratorum*, which prefer forests, edges, and transitional habitats, increased at lower elevations (Figure 6g), which may be associated with scrub encroachment and the expansion of tree boundaries fostered by climate warming [33,78]. Thus, our results support evidence from previous studies that climate and land use changes lead to a loss of specialized species and a dominance of generalists [40,82]. We could show similar trends for the climatic preferences of the community, as we observed a significant increase in an indiscriminate macro- (Figure 6k) and wide microclimatic preference (Figure 6i) that can be directly linked to temperature rise. As mentioned before, this is related to dominance of some generalist species that tend to be better adapted as they have greater tolerance toward climatic fluctuations and greater dispersal abilities [30,37]. We did not detect a significant increase in community weighted means of the species temperature index (Figure 6h) or a decrease in species preferring cold (Figure 6j) and alpine climates (Figure 6l). However, we suspect that this is likely influenced by the elevational resolution of the historical data, and we did not analyze species abundance because we lacked information on historical survey methods. The average spring temperature increased by 1.76 K (Figure 2), which corresponds to an elevational shift of the equivalent climatic habitat 257 m upwards. As the elevational uncertainty was between 200 and 500 m for many sampling units, the historical data source might not be sufficient to visualize upward shifts of bumblebees as shown in previous studies [11,39]. 

### 4.3. Community Changes in the Context of Range Shifts and Extinction Debts

The different response of specialist and generalist species to climate change can be related to different processes of range shifts. Range borders are defined by physiological constraints and competition, and across an ecological gradient-like elevation, the more stressful edge—in our case, the upper limit—is set predominantly by physiology and the lack of resistance mechanisms [2,83]. Contrarily, the low altitude edge is defined mainly by competition. When climate change leads to an upwards shift of potential limits, colonization by species released from the constraints can be quite fast, while the retreat of the weaker competitors might be delayed. The increase of plant species numbers in response to climate change in high altitude vegetation, which had been shown after comparison with the historical species list, was attributed to this effect [84]. The lack of increase of community weighted means of the species temperature index and occurrence of cold adapted species might be an expression of this, and the decline of species richness is therefore mainly related to the decrease of specialized species. General concepts hold that specialized species retreat or vanish faster than generalists with decreasing habitat quality (e.g., [85]). Our data indicate that the temperature increase in spring can shift the composition of the community and play a role in climate-driven range shifts. Our results show that specialized species, such as *B. ruderarius, B. monticola, B. mendax, B. bohemicus, B. sylvestris*, and *B. flavidus*, are less able to adapt to climate warming, and the potential ecological mechanical processes underlying this are related to temperatures increasing during their solitary phase in spring [28,79]. Contrarily, generalist species, such as *B. pascuorum, B. pratorum,* and *B. lucorum/cryptarum* complex, can better adapt and colonize new habitats due to their higher dispersal ability and broader climatic tolerance. This disparity in adaptation is especially affecting socio-parasitic bees, as their habitat quality depends on the population density of their host species, and recolonization could be impeded by low host density [22]. The observation of different behavior in retreat and colonization at the range edges suggests different extinction debt related lag times for specialist and generalist species [41]. This implies that the changes might continue and also increasingly affect generalist species that are not obviously influenced yet because the trigger for a future decline might have already occurred in the past (e.g., [42]). Comparisons like the one presented here, and the detailed continuation of monitoring in the future, promise to verify these scenarios and are therefore an important prerequisite to estimate the future development of biodiversity.

## 5. Conclusions


In summary, we have shown that changes in species diversity and functional composition of bumblebee communities in alpine ecosystems are related to climate warming, specifically to higher spring temperatures. Changes in distribution, phenology, and physiology are known adaptive mechanisms of species [43]. However, changes in distribution are only possible if species can keep pace with environmental changes and depend on the ability of species to colonize new climatically suitable areas and overcome landscape structures. Our study shows a significant functional shift toward generalists and a loss of species with specialized ecological preferences (Figure 6) that relate to increased temperatures in spring. Climate warming and land use changes lead to habitat degradation and consequently to a reduction and fragmentation of suitable areas and environmental conditions. It is likely that trends towards species loss will continue, especially at higher alpine elevations, and increasingly affect generalist species that are not obviously influenced yet, due to different extinction debt related lag times. As extreme climate events, such as drought and heavy rainfall, are expected to become more frequent, future studies could also examine the relationship between species composition and precipitation patterns. Based on our findings, we propose measures to protect climatically suitable and high-quality habitats: These include reducing usage competition with human activities, preserving open alpine pastures, and promoting extensive grassland rich in plant species. Our study also demonstrates the great potential of evaluating historical data sources as reference databases prior to global warming and modern land use changes. However, we also show their limitations, as the resolution of the historic samples is often not sufficient to answer all ecological questions of modern research in detail. To fully understand changes in bumblebee communities, it is necessary to conduct further surveys over a longer time period and consider other covariates such as land use and vegetation cover given that reliable data are available.

## Figures and Tables

**Figure 1 biology-12-00316-f001:**
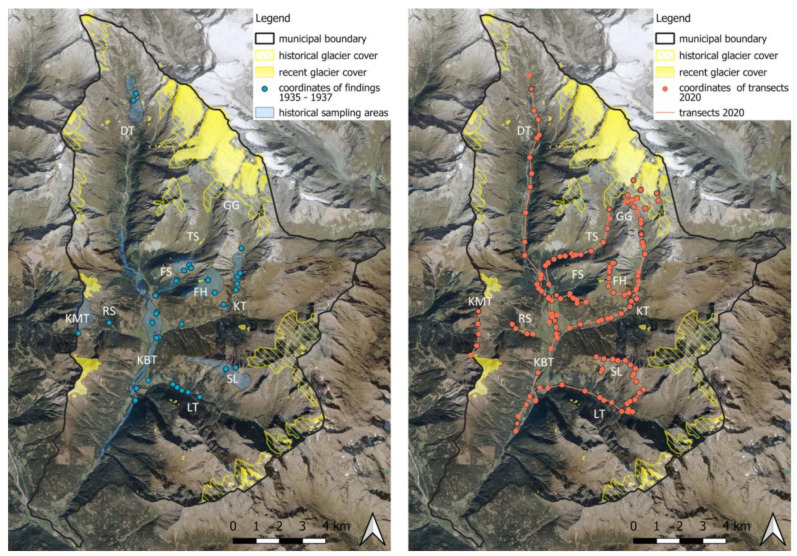
Sampling areas around Kals am Großglockner. The left plot shows the locations from 1935 and 1937 (blue dots) and the right plot the surveyed transects in 2020 (red dots and transects). Historical glacier cover is shown in yellow dashed areas [44], and recent glacier cover in yellow filled areas [50]. The abbreviations of the sampling sites are as follows: Kalsbachtal (KBT), Lesachtal (LT), Ködnitztal (KT), Foledischnitz (FS), Teischnitztal (TS), Dorfertal (DT), Raseggbachtal (RS), Kals-Matreier-Törl (KMT), Schönleitenspitze (SL), Figerhorn (FH), and base of the mountain Großglockner (GG). The municipal boundary is shown in black (data: community boundary—BEV, 2019, basemap.at, accessed on 5 January 2023).

**Figure 4 biology-12-00316-f004:**
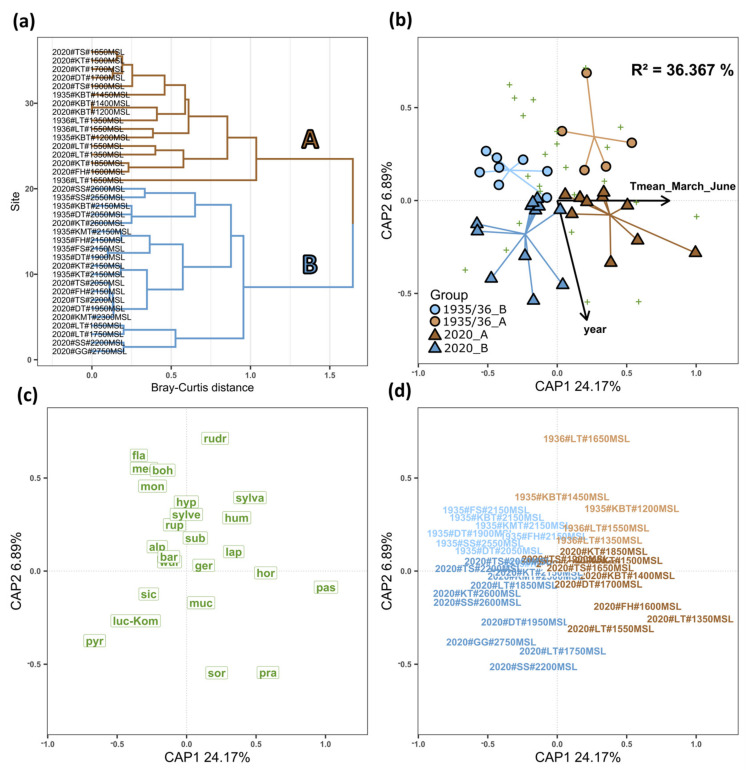
Analyses of bumblebee community composition of sampling units. (**a**) Cluster analysis (Hierarchical Clustering, method = “ward.D2”) and the clusters (A = brown and B = blue) extracted at Bray-Curtis distance = 1.5. (**b**) Ordination biplot of CAP modeling mean annual spring temperature and sampling period as constraints. Each point represents the community composition of one sampling unit, while colors and shapes indicate the four cluster groups (light colored circles = 1935/1936, dark colored triangles = 2020). Vectors indicate correlation of numeric values of mean annual spring temperature (“Tmean_March_June”) and sampling year (“year”) with community composition of sampling units. Values on axes refer to the percentage of explained variance. Name labels of sampling units, in reference to positions in (**b**), are shown in (**d**). Green crosses in (**b**) refer to the position of the species in the reduced ordination space, whose labels are shown in (**c**). Abbreviations in (**a**) and (**d**) denote sampling units: the first character string denotes sampling year (1935/1936 or 2020), second string denotes location (Kalsbachtal (KBT), Lesachtal (LT), Ködnitztal (KT), Foledischnitz (FS), Teischnitztal (TS), Dorfertal (DT), Raseggbachtal (RS), Kals-Matreier-Törl (KMT), Schönleitenspitze (SL), Figerhorn (FH), base of the mountain Großglockner (GG)), and third string denotes mean elevation of sampling unit (MSL = meters above sea level). Abbreviations in (**c**) are as follows: alp = *B. alpinus,* bar = *B. barbutellus,* boh = *B. bohemicus*, fla = *B. flavidus*, ger = *B. gerstaeckerii*, hor = *B. hortorum*, hum = *B. humilis*, hyp = *B. hypnorum,* jon = *B. jonellus*, lap = *B. lapidarius*, luc-Kom = *B. lucorum/cryptarum complex*, men = *B. mendax*, mon = *B. monticola*, muc = *B. mucidus*, pas = *B. pascuorum*, pra = *B. pratorum*, pyr = *B. pyrenaeus*, qua = *B. quadricolor*, rudr = *B. ruderarius*, rup = *B. rupestris*, sic = *B. sichelii*, sor = *B. soroeensis*, sub = *B. subterraneus*, sylva = *B. sylvarum*, sylve = *B. sylvestris*, wur = *B. wurflenii*.

**Figure 5 biology-12-00316-f005:**
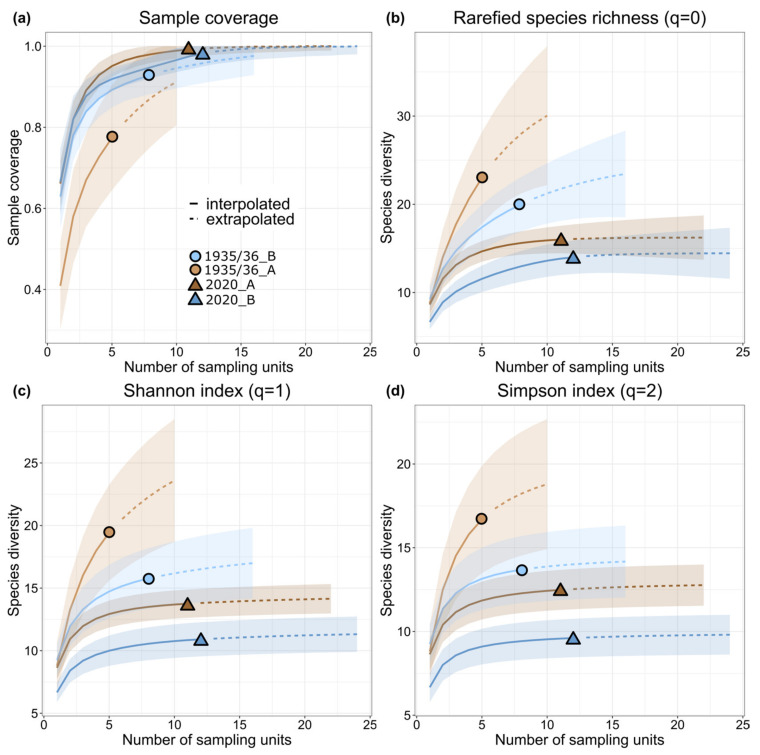
(**a**–**d**) Sample unit-based diversity estimates plotted as integrated curves for each sampling period (1935/1936 = circles, 2020 = triangles) and elevation clusters (A = brown, B = blue); (**a**) Sample coverage of each group and (**b**–**d**) three integrated diversity curves representing Hill numbers (q): the rarefied estimates (**b**) for species richness (q = 0), (**c**) for Shannon’s diversity index (q = 1), and (**d**) for the inverse Simpson’s index (q = 2). Colors and shapes represent the respective sampling period and elevation cluster. The interpolation estimates are represented by solid lines and the extrapolation estimates by dashed lines.

**Figure 6 biology-12-00316-f006:**
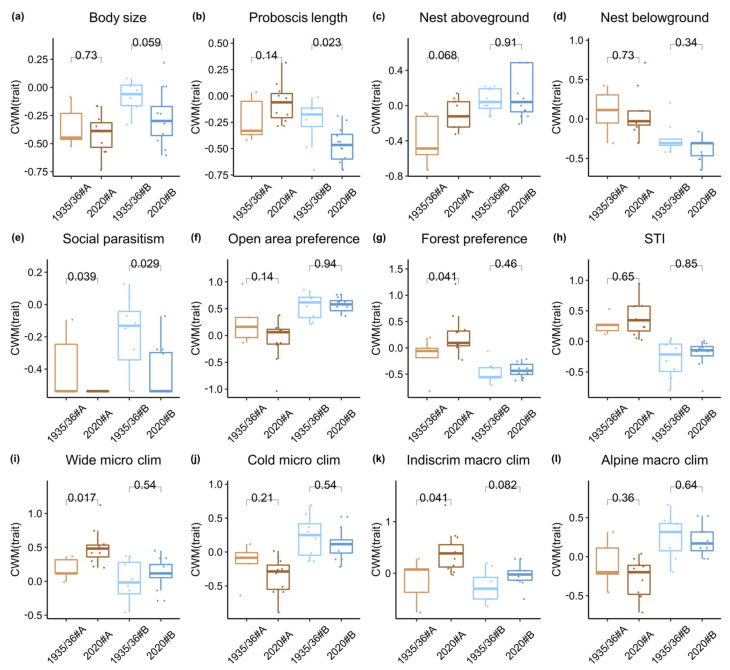
(**a**–**l**) Distribution of bumblebee species traits. For each cluster group, the CWM (community weighted mean) values of the (**a**) worker body size in millimeters, (**b**) worker proboscis length in millimeters, (**c**) share of aboveground nesting species, (**d**) share of belowground nesting species, (**e**) share of social parasitic species, (**f**) habitat preference for open areas, (**g**) habitat preference for forests, (**h**) a species temperature index (STI) according to Rasmont et al. [20], (**i**) wide microclimatic preference, (**j**) cold microclimatic preference, (**k**) indiscriminate macroclimatic preference, and (**l**) alpine macroclimatic preference are shown. Traits have been scaled and centered prior to CWM calculation, and y axes hence show transformed values. The first character string of cluster groups as shown on the x axis denotes the sampling year (1935/1936 or 2020), and the second denotes the group (mean elevation of 1550 masl (brown, A) and 2150 masl (blue, B)). Boxes represent CIs, lines represent means, and boxes represent +/− 1.5* (interquartile range). Dots represent actual values of sampling units. Letters indicate *p*-values of the Kruskal-Wallis rank sum test comparing the two sampling periods of the two cluster groups.

## Data Availability

The data and the corresponding R-script presented in this study are openly available in [FigShare] at [https://doi.org/10.6084/m9.figshare.21829326.v1, accessed on 5 January 2023].

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
