# Peer review of "Changes in Community Composition and Functional Traits of Bumblebees in an Alpine Ecosystem Relate to Climate Warming"

_biology, 2023, doi:10.3390/biology12020316_

Round 1

Reviewer 1 Report

Dear Authors,

You conducted interesting research that has the great importance for prediction of changes in bumblebee ranges under the impact of climate change. You wrote in the Materials and Methods that the Pittioni collections, among the historical bumblebee data, are one of the most accurate, which allowed you to conducted your research. Are there any other similar collections in other regions of Europe? If you have this type of information, is it possible to briefly mention it in the Introduction? In my opinion (I also study bumblebees), this information would be useful to the readers of your article, as well as for the future research on this topic (of course, if you have information about it).

In general, I have no fundamental corrections to your manuscript. However, I would recommend improving a little for a better reading of your article. You use a lot of abbreviations (there is nothing negative), but I would recommend not using them in the Discussion, because this makes it difficult in something degree to read the text. Reader needs to go back to the previous sections and looking for them. These are STI and CWM (lines 477, 491, 547, 566, 567).

In Figure 4 (especially here) and Figure 6, you did not give the full form for all the abbreviations that used here. I would recommend give them completely in captions, although they will increase a little the volume of the text.

Best wishes

Author Response

Dear Reviewer,

Thank you for taking the time to review our manuscript "Changes in community composition and functional traits of bumblebees in an alpine ecosystem relate to climate change" and for your valuable feedback.

  1. Regarding your suggestion to mention other similar collections in other regions of Europe in the Introduction: yes, for example, comparable collections exist in Norway where the records have been compiled from various sources and later digitized through Artskart (https://artskart.artsdatabanken.no/). We now included this information in the introduction of our article. We agree that it provides useful context for readers and supports future research on the topic of bumblebee ranges and climate change.
  2. We understand your point about the use of abbreviations in the Discussion section making it difficult to read the text. We now used full forms of the abbreviations STI and CWM in the Discussion section to improve readability.
  3. We apologize for not providing the full forms for all the abbreviations used in Figure 4 and Figure 6. We now included them in the captions to improve clarity for the readers.

Your constructive feedback has improved the revised version of our manuscript.

Best regards,

Victor Sebastian Scharnhorst (on behalf of all authors)

Reviewer 2 Report

It was very interesting to me to revise the manuscript entitled " Changes in community composition and functional traits of 2 bumblebees in an alpine ecosystem relate to climate warming" 

The work looks great with very good methodology and data interpretation. the writing flow was good. All data used were provided and the discussion section is near completeness

 Congratulation 

Author Response

Dear Reviewer,

Thank you for taking the time to review our manuscript "Changes in community composition and functional traits of bumblebees in an alpine ecosystem relate to climate warming" and for your positive feedback. We are pleased to hear that you found the work interesting and that you found the methodology and data interpretation to be good. We are also glad to hear that you found the writing flow to be good and that all data used were provided.

We appreciate your recognition of the near completeness of the discussion section, we will make sure to take into consideration your remark and improve it even more.

Best regards,

Victor Sebastian Scharnhorst (on behalf of all authors)

Reviewer 3 Report

Good manuscript; well written and organized. The changes in bumblebee community composition and functional traits are clear, and the elevated spring temperature is quite likely the factor. I have only a few questions about data, also some minor suggestions:

1. Based on L362-369, it appears that after variable selection the constrained ordination, which explains 36.367% of total variation, includes only two predictors (mean spring temperature and sampling year), while other environmental predictors, such as mean elevation, mean annual temperature, or mean winter temperature (October-March), are absent in CAP model. However, why mean elevation (MSL-mean) is also included in the biplot in Figure 4b (also see L379)? Interestingly in the figure MSL-mean appears to be very contributive in the variation explained by CAP1, because the MSL vector has a highly negative score in CAP1 and negatively correlated with mean spring temperature. Not surprising because MSL separates groups efficiently (Figure 4a) in the hierarchical cluster analysis (although unconstrained). Looks like there is an inconsistency to be addressed.

2. Does the CWM decrease in social parasitism in 2020 (Figure 6e) corresponds to the strong association between positions of cuckoo bees (Figure 4c) and 1935/36 group (Figure 4b)? If do I suggest highlight it somewhere in discussion.

 Others:

1. L17, 32: what species, animal and/or plants? Could be more specific.

2. L122: separated -> separately clustered

3. Suggestion: Use “masl (meters above sea level)” in L117 and keep using masl thereafter.

4. L274: such -> such as

5. L344: α = 0.01 or P < 0.01

6. L315: suggest list all cuckoo bee species, this is helpful for understanding figure 4c and 6e.

7. Figure 5: Attach the nonparametric K-W test results (p-value) in each panel.

8. Throughout the text use “Inverse Simpson index” to indicate the Hill number when q=2. Such as in Figure 5d, L418, L516.

9. L450: Proboscis length

10. L452: CMW -> community weighted mean (CMW) 

Author Response

Dear Reviewer,

Thank you for your constructive feedback on our manuscript. We appreciate the positive remarks and are pleased to see that our study on the changes in bumblebee community composition and functional traits is clear and well-organized.

Regarding your questions and suggestions:

  1. Thank you for bringing the inconsistency regarding the inclusion of mean elevation (MSL-mean) in the biplot of Figure 4b to our attention. We have addressed this issue by removing MSL-mean from the biplot. Our selection of predictors was based on the results of the constrained ordination, which only included mean spring temperature and sampling year, and found that mean spring temperature was significantly more contributive in the variation compared to mean elevation, even though both factors are highly correlated.

  2. We appreciate your suggestion to highlight the possible relationship between the decrease in social parasitism in 2020 and the strong association between positions of cuckoo bees and the 1935/36 group. We have added a brief discussion on this relationship in the revised manuscript.

Others:

  1. We apologize for the lack of specificity. We have specified the species in question as both invertebrates and plants.

  2. We have made the suggested change to "separately clustered" in reference to the species.

  3. We have used "masl (meters above sea level)" consistently throughout the manuscript.

  4. We have made the suggested change to "such as" in reference to the environmental variables.

  5. We have changed "α = 0.01" to "p-value < 0.01" to meet statistical standards.

  6. We have listed all cuckoo bee species in the revised manuscript to improve understanding of Figure 4c and 6e.

  7. We have considered the suggestion to attach the nonparametric K-W test results (p-value) to each panel in Figure 5, however, after careful consideration, we have decided not to include them in the figure. Our reason for this decision is that the addition of p-values to the plot would over-clutter the visual representation of the data and make it more difficult for the reader to quickly grasp the main results and trends. Instead, we have chosen to report the p-values in the corresponding text only, where they can be easily referenced and discussed in greater detail. This approach allows us to present the data in a clear and concise manner, while still providing the necessary information for readers to interpret and understand the results.

  8. We have used "Inverse Simpson index" to indicate the Hill number when q=2 throughout the text, as suggested.

  9. We have made the suggested change to "Proboscis length."

  10. We have made the suggested change to "Community Weighted Mean (CMW)" in reference to the metric used to describe the functional trait of the bumblebee communities.

Once again, we appreciate your feedback and the opportunity to make these improvements to our manuscript.

Best regards,

Victor Scharnhorst (on behalf of all authors)

Reviewer 4 Report

The study focuses on the change of the bumblebee community as a result of climate change. However, there are other parameters such as land use, vegetation, extreme conditions such as fires that can also cause major changes. In many publications it is shown that many species move altitudinally to find their climatic niche, but reducing everything to the climatic factor can lead to error and reduce the impact of other variables. I would recommend that the authors also compare the type of vegetation or land use as they are variables clearly related to bumblebees to provide a broader vision.

On the other hand, the study data are reduced to 2 periods: 1935-1936 or 37 (please review the dates because depending on which paragraph appears one date or another) and 2020 (1 month). In this sense, long-term studies show that the abundance and number of species are variable according to the year and there are years with more abundance than other. Therefore, focusing all the results at 1 month (2020) of sampling is very fine spinning because many different variables can affect. To understand these changes, you should sample at least one or two years more to understand the communities properly. 

Other points to comment are:

- Line 156: there are data from other years? This would allow us to understand better the changes.

- Line 167: How long does each transect? Is the same time used in each one?

- Lines 167: At what time of day was sampling? Is it done at different day times to check the biodiversity of bumblebees? Previous observations say that each species has different time preferences.

- Line 177: how many individuals or type species were excluded? Perhaps it is an interesting point to add to the models.

- Line 193: Sampling was carried out between July and August 2020. Why are these months not included in the temperature analysis?

- Line 463: The reproduction of methods and sampling times are very important when comparing data and moments. As you mentioned in line 156 sampling strategy of Pittioni is unknown, but we know that at this time the samples were made by observation and sweeping net so you replicated it very well and, in addition, you describe clearly the methodology to be able to replicate it in the future. Well done!

Author Response

Dear Reviewer,

Thank you for taking the time to review our manuscript and for providing valuable feedback. We appreciate your comments and have taken them into consideration to improve our study.

In response to your points:

  1. The impact of land use and vegetation on bumblebees: You are correct that factors like land use, vegetation, and extreme events can also impact bumblebee communities. Our study primarily focused on the relationship between spring temperature and bumblebee community composition, given the data available and previous research findings. However, the lack of reliable data on land use and vegetation cover during the 1935/36 reference period is acknowledged as a limitation. In the revised manuscript, we address this issue in the discussion and highlight the importance of considering these variables in future studies, if data are available.

  2. Data period: We appreciate your concern regarding the data period. Our study focuses on the comparison between bumblebee communities from 1935/1936 and 2020, reflecting a significant time interval of climate change impact. Although we acknowledge the importance of long-term studies on bumblebee communities, we have included a thorough evaluation of the limitations of one- or two-year surveys in our revised manuscript. It's also crucial to note that Pittioni did not sample the same locations in both 1935/1936, but rather different locations in those years, hence historical sampling did not cover the same locations over two years. We have now highlighted this information in the revised manuscript.

  3. Data from other years: We did not have histroical data from years other than 1935 and 1936. Only a few scattered records are available from 1937, and it can be assumed that Pittioni did not conduct extensive sampling after 1936. Hence, we excluded the 1937 data from our analysis. We have added a sentence in the revised methods section to clarify this point. However, we will consider collecting data from more years in future studies in the same area.

  4. In regards to the sampling times and transect length: we conducted our surveys throughout the day as some areas required the entire day to be mapped due to their size and it was not feasible to stop and resume the next day. However, when possible or when switching to a different area, we tried to avoid the mid-day "downtime" when fewer bumblebees were active. Our study does not provide any definitive insights into the time preferences of different bumblebee species. Any statements in this regard would be based solely on subjective observations and experiences, as there is currently no established research on this topic. Our main mapping time was between 9am and 5pm, with some activity still observed until 6pm in some areas. Our approach was generally based on the environmental conditions rather than the specific time of day. We have specified our approach in the revised version of our manuscript

  5. Excluded individuals or species: We excluded individuals or species that could not be accurately identified due to difficulties in capturing them in the field. A small number of bumblebee specimens were recorded as Bombus sp. due to being either uncatchable or flying away. These specimens were omitted from any subsequent analysis. However, all specimens that were captured and transferred to the lab for identification were successfully identified to species level. This has been clarified in the revised version of the manuscript. After careful consideration of your comment we decided not to include these specimens as they would add uncertainty and bias to the data, which would negatively impact the conclusions drawn from the analysis. By excluding these uncatched and thus undetermined specimens, we ensure that the results are based on accurate and reliable data, which is important for the validity of the conclusions and the overall quality of the study.

  6. Sampling months: Our analysis aimed to determine changes in species composition over a longer period of time and thus, the climatic input variables were selected to reflect that. We used the 30 years prior to the sampling period to get a robust estimate of the climatic conditions that determine the long-term composition of the habitat. We have further clarified our approach in the revised method section of our manuscript. It should be noted that while the current weather might influence which species are caught on the sampling day, this effect cannot be explained by our models and is considered a small effect.

    Regarding the specific months of July and August 2020, we do not have information about the historical weather conditions during the historical sampling days that would be essential for comparison. However, it can be assumed that the sampler at that time, Pittioni, also took advantage of sampling in "good" conditions (as we did) during the whole day. Additionally, reaching certain altitude levels requires the use of the whole day, as it is practically not possible to reach these levels otherwise. There is a small bias as a result of starting at lower altitudes in the morning and collecting higher altitudes from late morning to afternoon, but this corresponds relatively well to the later warming and earlier afternoon cooling higher up. Please see also our response above on your comment in regards to the sampling times and transect length. We plan to consider including a comparison between the effects of weather and long-term climate data in future temperature analyses - thanks for the hint.

  7. Reproduction of methods and sampling times: Thanks for your comment. We appreciate your recognition of our efforts to replicate the sampling strategy of Pittioni, despite the lack of information about the exact methods used. Our aim was to provide a clear and detailed description of the methods used in our study to ensure replicability in the future. We are glad that our efforts have been successful in this regard. 

Thank you again for your thoughtful feedback.  We believe that our manuscript has been improved as a result.

Victor Scharnhorst (on behalf of all authors)

Round 2

Reviewer 1 Report

Dear authors,

Thank you for taking into account my recommendations to your manuscript.

Best wishes

Reviewer 3 Report

Dear authors: The manuscript has been sufficiently improved. Only a few small corrections in the final proofreading.   1. The title of Figure 6b: Proboscislength -> Proboscis length 2. "Bombus" in italics: L336, L1226, L1228, L1242 ("Bombus alpinus"), L1345, L1353 3. L772: Proboscis -> proboscis     Best wishes

Reviewer 4 Report

Dear authors,

Thanks for your changes. 

Kind regards.